# Distribution patterns of Acidobacteriota in different fynbos soils

**Tersia Andrea Conradie**🆔⍟, **Karin Jacobs**🆔⍟*

Department of Microbiology, Stellenbosch University, Stellenbosch, Western Cape, South Africa

⍟ These authors contributed equally to this work.
* kj@sun.ac.za

**Data Availability Statement:** The data underlying the results presented in the study are available from GenBank with accession number PRJNA682354 (www.ncbi.nlm.nih.gov/bioproject/PRJNA682354).

## Abstract

The Acidobacteriota is ubiquitous and is considered as one of the major bacterial phyla in soils. The current taxonomic classifications of this phylum are divided into 15 class-level subdivisions (SDs), with only 5 of these SDs containing cultured and fully described species. Within the fynbos biome, the Acidobacteriota has been reported as one of the dominant bacterial phyla, with relative abundances ranging between 4–26%. However, none of these studies reported on the specific distribution and diversity of the Acidobacteriota within these soils. Therefore, in this study we aimed to first determine the relative abundance and diversity of the Acidobacteriota in three pristine fynbos nature reserve soils, and secondly, whether differences in the acidobacterial composition can be attributed to environmental factors, such as soil abiotic properties. A total of 27 soil samples were collected at three nature reserves, namely Jonkershoek, Hottentots Holland, and Kogelberg. The variable V4-V5 region of the 16S rRNA gene was sequenced using the Ion Torrent S5 platform. The mean relative abundance of the Acidobacteriota were 9.02% for Jonkershoek, 14.91% for Kogelberg, and most significantly ($p<0.05$), 18.42% for Hottentots Holland. A total of 33 acidobacterial operational taxonomic units (OTUs) were identified. The dominant subdivisions identified in all samples included SDs 1, 2, and 3. Significant differences were observed in the distribution and composition of these OTUs between nature reserves. The SD1 were negatively correlated to soil pH, hydrogen ($H^+$), potassium ($K^+$) and carbon (C). In contrast, SD2, was positively correlated to soil pH, phosphorus (P), and $K^+$, and unclassified members of SD3 was positively correlated to $H^+$, K, and C. This study is the first to report on the specific acidobacterial distribution in pristine fynbos soils in South Africa.

## Introduction

The Acidobacteriota are considered as one of the most ubiquitous and highly abundant soil bacteria. This phylum was first described in 1997 with only three cultured representatives [1]. Currently, the Acidobacteriota is divided into 15 class-level subdivisions, of which only 5 subdivisions contain the 62 successfully cultured and fully described species of the Acidobacteriota [2, 3]. The application of 16S rRNA gene sequencing techniques has revealed that this phylum can represent up to 50% of the total bacterial community, averaging around 10–20% of the

**Funding:** This research was funded by the National Research Foundation of South Africa (www.nrf.ac.za), grant number 112070. The funders had no role in study design, data collection and analysis, decision to publish, or preparation of the manuscript.

**Competing interests:** The authors have declared that no competing interests exist.

global soil bacteria [4–7]. Environmental surveys of different biomes indicated that the distribution of the Acidobacteriota is not limited to the soil environment, but can be found in several marine habitats, as well as some extreme habitats, such as acid mine drainage and hot springs [4, 8, 9]. Their prevalence in soils suggest that the Acidobacteriota play an important role in biogeochemical processes, as microorganisms are an essential part of the terrestrial environment and are important in maintaining ecological functions [10]. These functions are especially important in biomes where nutrient availability is low, and plants depend on their symbiotic relationships with the soil microbiome. One example of such a biome includes the fynbos biome in the Cape Floristic Region (CFR) of South Africa [11, 12]. Soils in the CFR are naturally low in nutrients, specifically in nitrogen and phosphorus, and acidic in nature [13, 14]. Fynbos plants have adapted to survive in the CFR, and the most significant contributing factor to their persistence in this hostile environment, is their symbiotic relationship with soil inhabiting microorganisms. These symbiotic interactions improve nutrient acquisition for these plants [11, 12].

In most high-throughput sequencing studies the effect of environmental factors, such as soil abiotic properties, for the most part is only studied at phylum, or at best class level [15–17]. However, these observations do not necessarily hold true at lower taxonomic classifications, as it assumes homogenous characteristics and that all species of the same lineage responds in a similar manner. One example is the correlation of Acidobacteriota with soil pH, the most prominent environmental factor that drives bacterial biodiversity [17, 18]. Several studies have indicated a stronger phylogenetic clustering of the Acidobacteriota as the pH decreases from neutrality [18–20]. However, not all Acidobacteriota are acidophilic, and the relative abundance of certain subdivisions, or even taxa within the same subdivision, could have a positive correlation with soil pH [20–22]. Other environmental factors could also be related to shifts in acidobacterial community compositions, such as soil organic carbon, phosphorus, and calcium [21, 23–25].

So far, studies on the fynbos biome only focused on the bacterial community as a whole and none have reported on the specific distribution of the Acidobacteriota within these soils or how environmental factors affect the acidobacterial composition [15, 26–29]. All previous knowledge of Acidobacteriota diversity in fynbos soils report a relative abundance of between 4–26%, with subdivisions 1 and 3 indicated as the major subdivisions [15, 26–29]. In this study, acidobacterial community diversity and composition were compared between fynbos habitats that are pristine, sampled from three fynbos nature reserves. Specifically, we aimed to first determine the relative abundance and diversity of the Acidobacteriota in three fynbos nature reserve soils, and secondly, whether differences in the acidobacterial composition can be attributed to environmental factors, such as soil abiotic properties.

## Materials and methods

### Experimental sites and sample collection

The collection of soil samples was approved by the Western CapeNature Conservation Board (permit: CN32-31-7035). Three sampling sites were selected based on their geographical location and fynbos types, namely Jonkershoek, Hottentots Holland, and Kogelberg (data in S1 Table, S1 Fig). The Jonkershoek Nature Reserve lies near the town of Stellenbosch and functions as a mountain catchment area for water supply to the town. The Hottentots Holland Nature Reserve lies on the opposite side of the Jonkershoek Mountains and on the northern stretch of the Hottentots Holland Mountain range. The Hottentots Holland Nature Reserve has an important role in the conservation of mountain fynbos. This area has some transformed areas (pine plantations, cultivation, and informal settlements), but remains mostly pristine

within the nature reserve. The Kogelberg Nature Reserve, also called the heart of the Cape Floristic Kingdom, lies on the southern stretch of the Hottentots Holland mountain range, and contains pristine protected mountain fynbos due to its internationally accepted conservation principles, guidelines and policies [30]. All three reserves have a main river along the hiking trails where all soil samples were collected, namely the Eerste River in Jonkershoek, and the Palmiet River in Hottentots Holland and Kogelberg.

Each site was sampled during the spring season (September-October) in 2019, after the cold, wet winter season typical of the fynbos biome [31]. In total, 27 bulk soil samples were collected, 10 at Jonkershoek, 8 at Hottentots Holland, and 9 at Kogelberg. Each sample collected was treated as a replicate for the different nature reserves. The organic layer was removed up to a depth of 10 cm and approximately 400 g of soil was collected from each sample site. The soil samples were kept on ice until processed. All samples were processed for DNA extractions within 24 hours after collection.

## Abiotic soil properties

Soil properties can vary considerably in South Africa's fynbos vegetation [32]. For this reason, soil abiotic properties were measured to capture any changes in these properties. The soil abiotic properties that were measured included soil pH, hydrogen ($H^+$), phosphorus (P), potassium ($K^+$), carbon (C), as well as extractable cations (EC) and base saturations (bs) of $Mg^{2+}$, $Na^+$, $Ca^{2+}$, and $K^+$. The analyses for the soil abiotic properties were conducted at BemLab (Somerset West, South Africa), using standard quality control procedures [33]. In short, air dried soil samples were sieved to remove any organic debris and plant roots. The pH of the soil was measured in a 1.0 M KCl solution (ratio of liquid to soil, 1:1) using an Ohaus Starter 2100 bench pH meter (Ohaus, Switzerland). Phosphorus (Bray II) and extractable cations concentrations (extracted with 0.2 M ammonium acetate at a pH of 7) were determined with ICP-OES analysis. Total C was measured by dry combustion with a Leco Truspec ® CHN analyser (SEAL Analytical Ltd., USA).

## Soil DNA extraction and sequencing

DNA of soil samples was extracted using the ZR *Quick*-DNA Fecal/Soil Microbe Kits (Zymo Research, USA), according to the manufacturer's instructions. PCR amplification of the successfully extracted DNA was performed with primers targeting the 16S rRNA V4–V5 region, 319-F (5′-ACT CCT ACG GGA GGC AGC AG-3′) and 783-R (5′-CTA CCA GGG TAT CTA ATC CTG-3′) [15]. The total reaction volume (25 μl) contained 1x KAPA Taq Hotstart Buffer, 1.5 mM $MgCl_2$, 0.5 μl of a 10 mM dNTP mix, 0.25 μM forward and reverse primers, 0.5 U KAPA Taq HotStart DNA Polymerase, and 1.5 μl of template DNA. PCR amplifications were performed on a 2720 Thermal Cycler (Thermo Fisher Scientific, USA) under the following conditions: initial denaturing at 95˚C for 5 minutes, followed by 35 cycles of 95˚C for 30 seconds, 58˚C for 30 seconds and 72˚C for 1 minute. A final extension was completed at 72˚C for 1 minute and the PCR samples were held at 4˚C. Concentrations and sizes of the PCR products were verified using the LabChip GX Software, v.5.4 (PerkinElmer Inc., UK). Samples were loaded onto an Ion 530™ Chip for sequencing using the Ion S5 Sequencing Systems (Ion Torrent, Thermo Fisher Scientific, USA).

## Sequence processing

The raw sequence data was submitted to GenBank as BAM files with accession number PRJNA682354. After conversion to FASTQ files, the raw sequence data were analysed using MOTHUR (v.1.44.3), following the tutorial available at http://www.mothur.org/wiki/454_

SOP, with some modifications [34, 35]. In short, trimmed and quality filtered sequences were aligned against the SILVA v.132 bacterial reference database released in December, 2017 (http://www.arb-silva.de/). Sequences with errors and chimeric sequences were removed with the pre.cluster and chimera.uchime [36] commands, respectively, and classified using a cut-off value of 80. Sequences were clustered into operational taxonomic units (OTUs) and normalised to 13659 sequences. Acidobacterial sequences were retrieved using the get.lineage command in MOTHUR for further analyses.

## Statistical analysis

All statistical analyses were performed in MOTHUR (v.1.44.3) and R (v.4.0.3, R Core Team 2013) using the Vegan package. Non-parametric Kruskal-Wallis H-tests were calculated for all abiotic soil properties, acidobacterial relative abundances, observed OTUs, and Shannon diversity indices between the three nature reserves. To determine if any OTUs were significantly differentially represented between nature reserves, the metastats command in MOTHUR was used. Non-metric multidimensional scaling (NMDS) plots were drawn in R using the Bray-Curtis distance matrix generated in MOTHUR. Further statistical evaluations of the NMDS plots were evaluated using Analysis of Molecular Variance (AMOVA). Spearman correlation tests were performed between soil abiotic properties and the sequence abundances of different subdivisions, as well as correlations between significant soil abiotic properties and NMDS ordinations, using 9999 permutations. For all statistical evaluations, a $p$-Value of 0.05 were considered as significant. Corrections for multiple comparisons were made using the Bonferroni method.

## Results

### Soil abiotic properties of different nature reserves

The selected soil abiotic properties and their mean values (± standard deviation) are summarised in Table 1. A significant difference ($p<0.05$) was observed between all soil abiotic properties measured, except for Na.bs (sodium base saturation). Predominantly, the Jonkershoek soil contained higher concentrations of all soil abiotic properties measured, followed by Kogelberg, and Hottentots Holland. The most pronounced difference was observed with regards to phosphorus (P) and potassium ($K^+$), where Jonkershoek had more than double the concentrations measured for Kogelberg and Hottentots Holland. The mean soil pH values and total carbon content measured is consistent with the acidic and oligotrophic properties of fynbos soils.

### Acidobacteriota community composition and Alpha-diversity

After quality filtering and removal of chimeras, a combined total of 796,584 partial 16S rRNA gene sequences with a mean amplicon length of 410 bp were obtained (Table 2). Of these, a total of 104,426 (~13.11%) were Acidobacteriota-affiliated reads, with relative abundances ranging between 3.43–30.32% in the different samples from Jonkershoek, Hottentots Holland, and Kogelberg (data in S2 Table). Other major taxa also identified in all nature reserve samples included the phyla Proteobacteria, Actinobacteria, Firmicutes, and Bacteroidetes (S2 Fig). The mean acidobacterial relative abundance was significantly higher ($p<0.05$) in the Hottentots Holland samples, with a mean relative abundance observed at 18.42%, followed by Kogelberg (14.91%) and Jonkershoek (9.02%). Overall, the mean Shannon diversity indices and observed OTUs were similar and ranged between 2.04–2.19, and 23.67–25.38, respectively.

The OTUs were classified into 33 phylotypes, representing members from various subdivisions (SDs) (Table 3). Fig 1 illustrates the major Acidobacteriota community composition

**Table 1. Soil abiotic properties for each nature reserve.**

| Soil properties | Jonkershoek | Hottentots Holland | Kogelberg | *p*-Value |
|---|---|---|---|---|
| pH | 4.17 ± 0.05 a | 3.41 ± 0.26 b | 3.53 ± 0.28 b | *** |
| H$^+$ (cmol/kg) | 3.39 ± 0.42 a | 2.92 ± 0.73 ab | 2.75 ± 0.68 b | * |
| P (mg/kg) | 9.01 ± 7.24 a | 1.80 ± 1.07 b | 2.67 ± 0.85 b | ** |
| K (mg/kg) | 89.20 ± 3.03 a | 21.00 ± 3.50 c | 41.90 ± 17.20 b | *** |
| C (%) | 4.19 ± 0.23 a | 3.16 ± 0.65 b | 3.39 ± 0.71 b | ** |
| Mg.(EC) (cmol(+)/kg) | 0.50 ± 0.09 a | 0.35 ± 0.10 b | 0.91 ± 0.38 a | ** |
| Na.(EC) (cmol(+)/kg) | 0.16 ± 0.04 a | 0.09 ± 0.02 b | 0.15 ± 0.06 a | ** |
| Ca.(EC) (cmol(+)/kg) | 0.98 ± 0.39 a | 0.50 ± 0.03 b | 1.46 ± 0.84 a | *** |
| K.(EC) (cmol(+)/kg) | 0.23 ± 0.01 a | 0.05 ± 0.01 b | 0.11 ± 0.04 b | *** |
| Mg.bs. (%) | 9.62 ± 2.35 b | 8.91 ± 1.71 b | 16.60 ± 3.01 a | *** |
| Na.bs. (%) | 3.04 ± 0.33 | 2.43 ± 0.81 | 2.84 ± 0.57 | NS |
| Ca.bs. (%) | 18.10 ± 4.85 ab | 13.30 ± 3.20 b | 25.20 ± 6.40 a | *** |
| K.bs. (%) | 4.46 ± 0.77 a | 1.32 ± 0.06 c | 2.00 ± 0.74 b | *** |

A significant difference is observed at: * $p<0.05$; ** $p<0.01$; *** $p<0.001$. NS, not significant.

Letters a-c indicates where the significance lies.

EC–exchangeable cations.

bs–base saturation.

Values are means ± standard deviation (n = 10 for Jonkershoek, n = 8 for Hottentots Holland, n = 9 for Kogelberg).

based on 16S rRNA gene sequence analysis. The different OTUs were grouped according to their respective SDs. Subdivisions 1 and 3 were further divided into either classified or unclassified. The majority of the OTUs identified belong to SDs 1, 2, and 3 (Fig 1). In the Kogelberg samples, SD1 represents up to 55% of the acidobacterial community, followed by SD3 (36%) and SD2 (7%). In both the Jonkershoek and Hottentots Holland samples, SD3 had the highest relative abundance, with 44% and 47%, respectively, followed by SD1 (42% and 45%, respectively), and SD2 (12.5% and 6.7%, respectively). Other SDs also identified but with relative abundances of less than 1%, included SDs 4, 5, 8, 11, 12, 13, 15, and an unclassified SD. Representative of SD1 included taxonomically known genera, such as *Acidipila*, *Occallatibacter*, *Candidatus* Koribacter, *Edaphobacter*, *Granulicella*, *Terracidiphilus*, *Acidicapsa*, *Bryocella*, *Telmatobacter*, *Acidobacterium*, and *Terriglobus*, and some representatives of SD3, *Bryobacter*, *Candidatus* Solibacter, and *Paludibaculum*. However, the majority of the OTUs identified are of unclassified, or not-yet-described, representatives of the Acidobacteriota and are, therefore,

**Table 2. Sequence results of all three nature reserves and their relative abundances, Shannon diversity indices, and observed OTUs.**

| Nature Reserve | Filtered reads[1] | Acidobacteriota Reads | Relative abundance | | Shannon Diversity Index | Observed OTUs |
|---|---|---|---|---|---|---|
| | | | (%) | *p*-Value[2] | | |
| Jonkershoek | 329,133 | 29,460 | 9.02 ± 3.54 b | 0.004 | 2.19 ± 0.12 | 25.00 ± 3.27 |
| Hottentots Holland | 245,006 | 43,691 | 18.42 ± 6.22 a | | 2.04 ± 0.18 | 25.38 ± 2.20 |
| Kogelberg | 222,445 | 31,275 | 14.91 ± 5.63 ab | | 2.11 ± 0.14 | 23.67 ± 1.50 |
| **Total** | **796,584** | **104,426** | - | - | - | **33** |

[1]Filtered reads exclude low-quality reads and chimeras.

[2]A significant difference is observed at $p<0.05$. Letters a and b indicates where the significance lies.

Values are means ± standard deviation.

**Table 3. OTU classification and relative abundances between Jonkershoek, Hottentots Holland, and Kogelberg.**

| OTU | SD | Taxonomic classification | | Relative abundance (%)[2] | | |
|---|---|---|---|---|---|---|
| | | **Family** | **Genus** | **Jonkershoek** | **Hottentots Holland** | **Kogelberg** |
| OTU1 | 3 | Bryobacteraceae | *Bryobacter* | 21.99 ± 2.34 | 32.37 ± 3.57 | 23.05 ± 4.05 |
| OTU2 | 1 | Unclassified | Unclassified | *16.60 ± 1.23* | 15.05 ± 2.87 | *12.40 ± 1.57* |
| OTU3 | 3 | 'Solibacteraceae' | *Candidatus* Solibacter | *15.74 ± 1.32* | *10.67 ± 1.80* | *9.11 ± 1.77* |
| OTU4 | 1 | Acidobacteriaceae | *Acidipila* | *5.38 ± 1.75* | 9.77 ± 1.65 | *15.92 ± 3.61* |
| OTU5 | 1 | Acidobacteriaceae | Unclassified | *6.21 ± 1.12* | 9.49 ± 1.45 | *14.90 ± 3.23* |
| OTU6 | 2 | Unclassified | Unclassified | *12.53 ± 1.97* | *6.72 ± 1.72* | 7.02 ± 1.44 |
| OTU7 | 3 | 'Solibacteraceae' | Unclassified | 5.76 ± 0.63 | 4.41 ± 1.08 | 3.97 ± 0.62 |
| OTU8 | 1 | Unclassified | Unclassified | *6.24 ± 1.01* | *1.81 ± 0.42* | *2.39 ± 0.73* |
| OTU9 | 1 | Acidobacteriaceae | *Occallatibacter* | 1.17 ± 0.22 | 2.62 ± 1.21 | 4.19 ± 1.23 |
| OTU10 | 1 | Acidobacteriaceae | Unclassified | 2.54 ± 0.77 | 1.38 ± 0.59 | 0.85 ± 0.39 |
| OTU11 | 1 | Koribacteraceae | *Candidatus* Koribacter | 0.96 ± 0.27 | 1.70 ± 0.39 | 1.69 ± 0.35 |
| OTU12 | 1 | Unclassified | Unclassified | 1.02 ± 0.28 | 1.03 ± 0.39 | 0.92 ± 0.20 |
| OTU13 | 1 | Acidobacteriaceae | *Edaphobacter* | *0.63 ± 0.17* | **0.51 ± 0.08** | ***1.24 ± 0.17*** |
| OTUs with relative abundances < 1% | | | | | | |
| OTU14 | 1 | Acidobacteriaceae | *Granulicella* | *0.90 ± 0.18* | **0.56 ± 0.12** | ***0.23 ± 0.06*** |
| OTU15 | 8 | Thermoanaerobaculaceae | Unclassified | 0.70 ± 0.19 | 0.25 ± 0.13 | 0.29 ± 0.27 |
| OTU16 | 1 | Acidobacteriaceae | *Terracidiphilus* | 0.13 ± 0.04 | 0.41 ± 0.18 | 0.23 ± 0.05 |
| OTU17 | 1 | Acidobacteriaceae | *Acidicapsa* | 0.18 ± 0.07 | 0.24 ± 0.08 | 0.20 ± 0.08 |
| OTU18 | 1 | Acidobacteriaceae | *Bryocella* | 0.18 ± 0.06 | 0.16 ± 0.06 | 0.16 ± 0.06 |
| OTU19 | 5 | Unclassified | Unclassified | 0.29 ± 0.10 | 0.16 ± 0.07 | 0.20 ± 0.40 |
| OTU20 | 1 | Acidobacteriaceae | Unclassified | 0.10 ± 0.04 | 0.15 ± 0.06 | 0.31 ± 0.08 |
| OTU21 | 13 | Unclassified | Unclassified | *0.04 ± 0.03* | 0.23 ± 0.10 | *0.23 ± 0.06* |
| OTU22 | UC[1] | Unclassified | Unclassified | 0.18 ± 0.06 | 0.06 ± 0.02 | 0.09 ± 0.03 |
| OTU23 | 1 | Acidobacteriaceae | *Telmatobacter* | 0.09 ± 0.05 | 0.07 ± 0.04 | 0.18 ± 0.10 |
| OTU24 | 3 | 'Solibacteraceae' | Unclassified | 0.10 ± 0.03 | 0.09 ± 0.04 | 0.05 ± 0.02 |
| OTU25 | 15 | Unclassified | Unclassified | *0.10 ± 0.02* | *0.03 ± 0.02* | *0.03 ± 0.01* |
| OTU26 | 12 | Unclassified | Unclassified | 0.06 ± 0.04 | 0.04 ± 0.03 | 0.02 ± 0.02 |
| OTU27 | 3 | Bryobacteraceae | *Paludibaculum* | 0.13 ± 0.06 | 0.02 ± 0.02 | - |
| OTU28 | 1 | Unclassified | Unclassified | 0.03 ± 0.01 | - | 0.03 ± 0.03 |
| OTU29 | 1 | Acidobacteriaceae | *Acidobacterium* | - | - | 0.03 ± 0.03 |
| OTU30 | 11 | Unclassified | Unclassified | - | - | 0.07 ± 0.07 |
| OTU31 | 4 | Unclassified | Unclassified | 0.01 ± 0.01 | 0.02 ± 0.01 | 0.02 ± 0.01 |
| OTU32 | 1 | Acidobacteriaceae | *Terriglobus* | 0.01 ± 0.01 | 0.01 ± 0.01 | - |
| OTU33 | 8 | Unclassified | Unclassified | 0.01 ± 0.01 | - | 0.01 ± 0.01 |

[1] UC–Unclassified.

[2] Values are means ± SD.

A significant difference was observed at $p < 0.05$. Significance between groups are indicated as: Jonkershoek–Hottentots Holland (underlined); *Jonkershoek–Kogelberg* (italics); **Hottentots Holland–Kogelberg** (bold).

indicated as Unclassified. These unclassified OTUs represented between 40–52% of the relative abundances.

The distribution of several acidobacterial OTUs and their observed relative abundances were significantly different ($p < 0.05$) between the nature reserves (Table 3). For the most part, these differences were observed between Jonkershoek and Hottentots Holland, and Jonkershoek and Kogelberg. For example, the *Bryobacter* (OTU1; SD3) had a significantly higher

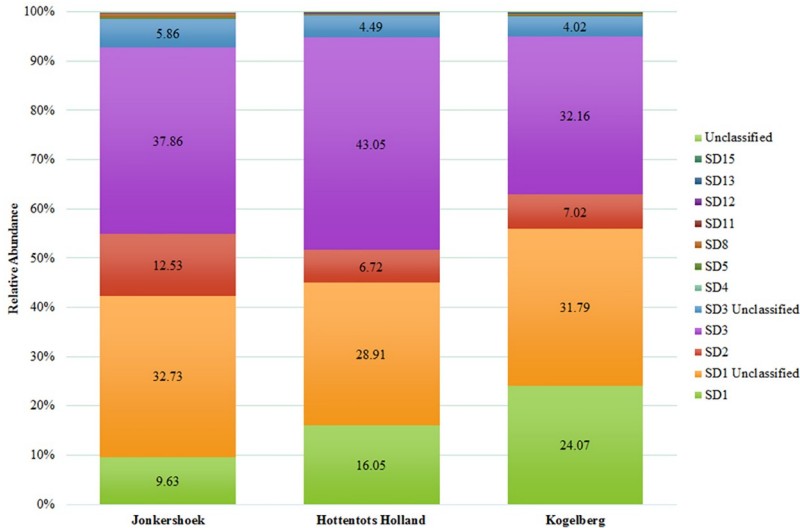

**Fig 1. The acidobacterial subdivision community composition based on 16S rRNA gene sequence analysis for Jonkershoek, Hottentots Holland, and Kogelberg.**

relative abundance in the Hottentots Holland samples (32.37%), compared to the Jonkershoek samples (21.99%). In the Jonkershoek samples, OTUs 3 (SD3), and 8 (unclassified SD1) had significantly higher relative abundances, compared to Hottentots Holland and Kogelberg. Several members of SD1 (OTUs 4, 5, 9, and 13) had significantly higher relative abundances in the Kogelberg samples, compared to the Jonkershoek samples.

## Beta-diversity

The non-metric multidimensional scaling (NMDS) revealed different acidobacterial community compositions corresponding to the different sites (Fig 2). The observed separation between the Jonkershoek and Hottentots Holland, as well as Jonkershoek and Kogelberg

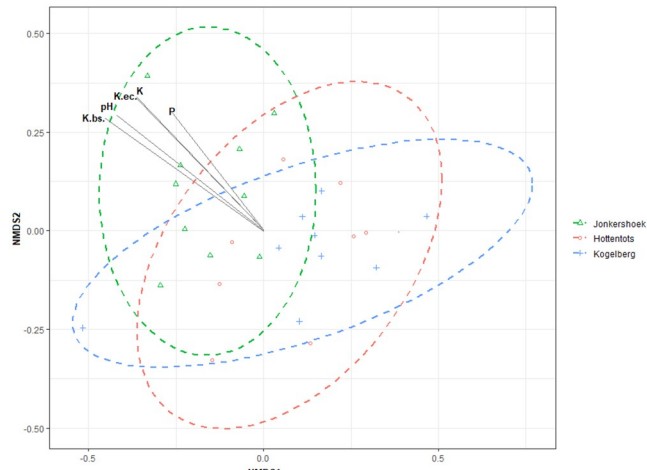

**Fig 2. A non-metric multidimensional scaling ordination plot, based on the Bray-Curtis distance matrix, representing the acidobacterial community compositions in various samples examined in this study.** Soil abiotic properties that had a significant correlation ($p<0.05$) with the acidobacterial communities are indicated with an overlaid bi-plot.

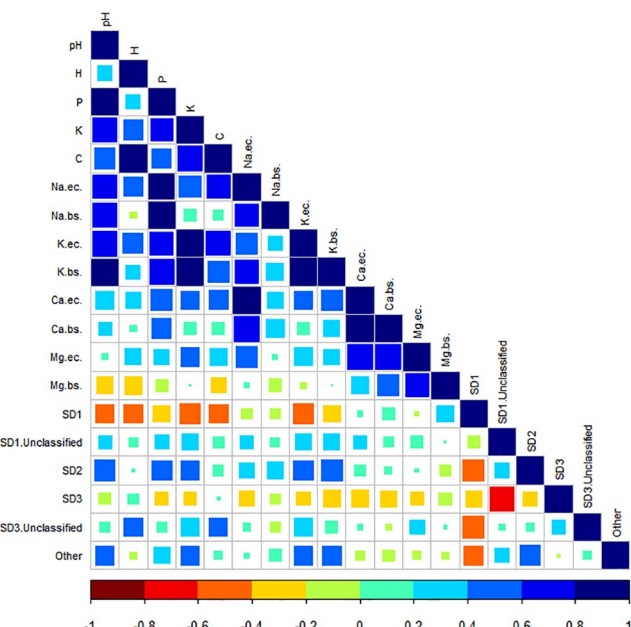

**Fig 3. Spearman correlations between acidobacterial subdivisions and soil abiotic properties.** Colour of the square indicates type of correlation (shades of blue–positive; green to dark red–negative). 'Other' contain subdivisions with relative abundances below 1%. SD1 Unclassified and SD3 Unclassified contain Acidobacteriota OTUs with no taxonomic classification at genus level. EC–exchangeable cations, bs–base saturation.

samples examined in this study was statistically significant with a *p*-Value of 0.021 and 0.008, respectively (data in S3 Table). Some soil abiotic properties responsible for the significant shift in acidobacterial community composition were identified as soil pH, phosphorus (P), and potassium (K, K.(EC), K.bs.). As these soil abiotic properties increased, the acidobacterial community shifted, with soil pH observed as having the greatest influence, $p<0.001$ (data in S4 Table).

Further evaluations of the effect of these soil abiotic properties on specific SDs and their relative abundances were examined using Spearman correlation tests (Fig 3, data in S5 Table). Significant correlations ($p<0.05$), positive and negative, were revealed for several soil abiotic properties and the relative abundances of acidobacterial subdivisions. For SD1 (Acidobacteriaceae), significant negative correlations were revealed for soil pH ($\rho$ = -0.404), $H^+$ ($\rho$ = -0.424), $K^+$ ($\rho$ = -0.475), C ($\rho$ = -0.432) and K.(EC) ($\rho$ = -0.488). As for SD2 and 'Other', positive correlations were revealed for soil pH ($\rho$ = 0.505; 0.438), P ($\rho$ = 0.485; 0.326), $K^+$ ($\rho$ = 0.435; 0.432), K.(EC) ($\rho$ = 0.463; 0.429), and K.bs ($\rho$ = 0.523; 0.433). Although 'Other' only had a weak positive correlation with P. Unclassified representatives of SD3 revealed a positive correlation with $H^+$ ($\rho$ = 0.430), $K^+$ ($\rho$ = 0.398), and C ($\rho$ = 0.408). Although statistical evaluations revealed these correlations to be significant, these correlations are only considered as moderate ($\rho$ = 0.4–0.6). Correlations can be considered as strong between $\rho$ = 0.6–0.7, and very strong at $\rho>0.8$.

## Discussion

In most high-throughput sequencing studies, differences at taxonomic levels between sample groups are traditionally only reported at phylum, or at best class level [15–17]. However, these observations ignore the fact that not all species of the same lineage responds in a similar manner, and valuable information is, therefore, lost within these assumptions. In this study we

have examined the Acidobacteriota community composition and distribution between three different nature reserves and how differences in the soil abiotic properties contributed to the observed shift in subdivision distribution. Previous studies on fynbos soils only reported on the microbial diversity as a whole and indicated Proteobacteria, Actinobacteria, Firmicutes, and Acidobacteriota as the dominant phyla present within these soils [15, 26–28, 37].

All previous knowledge of Acidobacteriota diversity in fynbos soils report a relative abundance of between 4–26%, and all of these studies report SDs 1 and 3 as the major subdivisions present within these soils [15, 26–28]. The results obtained in this study for fynbos nature reserve soils agree with these previous studies. The mean relative abundance of the Acidobacteriota were the highest in the sandy soils of Hottentots Holland (18.42%) and Kogelberg (14.91%), and lowest in the clay-loamy soil of Jonkershoek (9.02%) (Table 2). The major SDs identified in these soils were SDs 1, 2, and 3 (Fig 1, Table 3). These SDs are also among the most commonly found Acidobacteriota subdivisions in other soil environments [3, 38, 39]. In previous fynbos studies, the presence of SD2 was not reported. However, their existence within this biome might have gone unnoticed when examining the bacterial community as a whole, as this SD had a mean relative abundance of between 6.72–12.53% from acidobacterial affiliated sequences (Fig 1). The highest relative abundance of this SD was observed in the Jonkershoek samples, which had significantly higher nutrient availability (Table 1). Previously, SD2 has for the most part only been reported in colder subarctic soils and currently contain no cultured representatives [22, 40, 41]. Their reported presence within the fynbos soils, where temperatures can rise well above 30°C during the dry, warm summers [42], indicate that they are capable of tolerating higher temperatures than the cooler subarctic soils. Another study focussing on the soil acidobacterial distributions between savannah-like Cerrado and Atlantic forest Brazilian biomes, also indicated a high relative abundance of SD2 in these warmer climates [43]. These observations are supported by a previous study that focused on the active bacterial community at different nutrient and temperature cycles of Arctic tundra [40]. They found that different members of SD2 were more abundant either toward the late summer, or early-summer periods, or even more so directly after spring thaw when substrates, such as organic carbon and nitrogen, are abundant and easily accessible. In Mediterranean climate systems, such as the fynbos biome, water availability fluctuates over time and soil microbial activity is strongly influenced by seasonal changes [15, 44, 45]. During wetter seasons, the rewetting of soil triggers strong microbial activity which increases the mineralisation of nutrient sources that have accumulated during the dryer seasons [45–47]. A sudden change in osmotic potential may also lead to cell lysis, resulting in the release of intracellular solutes that are now available [46, 48]. Ivanova et al. [22], have also reported on the positive correlation of the SD2 relative abundance with soil carbon and nitrogen [22]. In this study, soil samples were collected in the early spring period after abundant rainfall during the wet, winter season in 2019. The observed increase in relative abundance of the SD2 in soils containing relatively higher nutrient concentrations, and its ability to tolerate a broad temperature range, from cooler subarctic to the warmer Mediterranean and Brazilian climates, are interesting characteristics of this SD. Future efforts in isolating members of this subdivision should consider these characteristics as part of their isolation strategy.

As revealed in this study, fynbos soils contain several members of classified and unclassified acidobacterial sequences, based on the 16S rRNA gene SILVA v.132 classifications (Table 3). Some of the major classified genera identified with relative abundances of >1%, included from SD1 *Acidipila*, *Occallatibacter*, *Candidatus* Koribacter, and *Edaphobacter*, and from SD3 *Bryobacter*, and *Candidatus* Solibacter. Their persistence in fynbos soils could be explained by the physiological characteristics of isolated and described members of these genera. Representatives of the genera *Acidipila*, *Occallatibacter*, and *Edaphobacter* were previously isolated from forest soils, an acid mine drainage site (AMD), Namibian savannah, and grassland soils [49–

56]. They are all classified as acidophilic and can grow at temperatures ranging between 10–40˚C. Several members of the Acidobacteriaceae are capable of degrading complex biopolymers such as xylan, pectin and chitin [3], suggesting an important role in biogeochemical processes, especially in the fynbos biome where soil nutrient availability is low, but plants are able to thrive [13, 14]. Representatives of *Candidatus* Koribacter (SD1) and *Candidatus* Solibacter (SD3) were previously isolated from Australian pasture soils. However, very little information is available about their physiologies [5, 57, 58]. The only isolated representative of the genus *Bryobacter* was isolated from peat bogs, is capable of growth at temperatures between 4–33˚C and is acidotolerant with growth at pH of 4.5–7.2 [59]. The clear dominance of the genus *Bryobacter* in these fynbos soils with relative abundances observed between 21–32% (Table 3), is surprising due to pH levels of below 4.5 measured for these soils (Table 1). As there is currently only one described member of this genus, *Bryobacter aggregatus* [59], this observation indicates that other members may be capable of growth at lower pH levels.

The Acidobacteriota community composition was significantly different between the nature reserves examined in this study (Fig 2). This can be mostly explained by the soil abiotic properties of these soil samples (Table 1). These soil parameters might not affect the functional role of the Acidobacteriota, but can influence the community composition [60]. With an increase in soil pH, phosphorus (P), and potassium (K, K.(EC), and K.bs), the community composition shifted significantly (Fig 2). Several OTUs from SD1, SD2, as well as SD3 were markedly different between these samples. The trend observed for described members of the Acidobacteriaceae (SD1) (Fig 3, data in S5 Table), agrees with our current knowledge of their affinity for low pH soils and their oligotrophic nature [3, 17, 19, 38, 61]. The negative correlation of SD1 with soil P and $K^+$ is in agreement with a previous study [62]. In contrast, members of SD2, SD3 Unclassified and 'Other' (subdivisions with relative abundances below 1%) had significant positive correlations to soil pH and several soil abiotic properties, including hydrogen ($H^+$), phosphorus (P), potassium ($K^+$, K.(EC), K.bs), and even carbon (C). These findings are in stark contrast to previous observations of these subdivisions [22, 43, 62], however, Naether et al. [21] have reported similar observations for SD3 [21]. Our understanding of the effect of soil abiotic properties on the relative abundance of different Acidobacteriota subdivisions is far from complete, as reported studies indicate contrasting results for the same SD [19, 21, 22, 38, 61–63]. However, these insights might still be helpful for a specific environment in future cultivation strategies, as different members of Acidobacteriota might have different survival strategies. It is possible that the Acidobacteriota within these pristine fynbos soils have adapted to this environment and, therefore, we see these contrasting observations of responses to soil abiotic properties and soil pH.

Soil abiotic properties are important factors to consider when designing cultivation studies, as all the subdivisions contain undescribed taxa. Future efforts in culturing these bacteria could benefit from their perceived ability to thrive under nutrient limitations, the importance of nutrient compositions to include low concentrations of P and $K^+$, and to possibly include a wider range of isolation medium pH, as was seen in this study and some other studies [22, 38]. The relatively high occurrence of the Acidobacteriota in these fynbos soils appear to agree with their phenotypic characteristics. A difficult task still lies ahead, but with each environmental survey of the Acidobacteriota, we get a glimpse into their metabolic potential and we can design better experiments to isolate novel species.

## Conclusions

The Acidobacteriota is a fascinating phylum that has received considerable interest within the last decade. Their detection in acidic soils are common, and this was also observed in this

study. The relative abundance of the Acidobacteriota was significantly higher in the Hottentots Holland samples, compared to the Jonkershoek samples, and the community composition also differed significantly between the Jonkershoek and Hottentots Holland nature reserves, and the Jonkershoek and Kogelberg nature reserves. The major subdivisions detected in these fynbos nature reserve soils, were SDs 1, 2, and 3. The relative abundance of SD2 was significantly higher in the Jonkershoek samples, possibly due to the higher nutrient availability. The relative abundance of some classified members of SD1 was significantly higher in Kogelberg, owing to their acidic and oligotrophic characteristics, whereas the relative abundance of some unclassified members of SD1 were significantly higher in Jonkershoek. Some interesting results were revealed with the observed positive correlation of SD2 and unclassified members of SD3 with several soil abiotic properties, including P, $K^+$, $H^+$, and C. Finally, a large percentage of sequences belonged to unclassified members of the Acidobacteriota. Without cultured representatives it becomes increasingly difficult to speculate on the functional role of the Acidobacteriota in fynbos soils. Their high relative abundance in nutrient poor fynbos soils, however, suggest that they play an important ecological role in biogeochemical cycles.

## Supporting information

**S1 Fig. Different geographical locations of the three nature reserves.** A. Hottentots Holland Nature Reserve, B-C. Jonkershoek Nature Reserve, and D. Kogelberg Nature Reserve.
(TIF)

**S2 Fig. Heat tree indicating the major taxonomic representations in the nature reserve soils.** Node sizes indicate OTU count, with larger nodes representing greater OTU representation. Colours Green to Pink indicate the representation of each node in the different samples. Green–Detected in all samples. Pink–Detected in only one sample.
(TIF)

**S1 Table. Sampling sites of three nature reserves and their fynbos and soil characteristics.**
(PDF)

**S2 Table. Summary of the filtered and Acidobacteriota-affiliated reads, together with the Acidobacteriota relative abundance for each sample ID.**
(PDF)

**S3 Table. Analysis of molecular variance (AMOVA) statistical results of comparisons between nature reserves.**
(PDF)

**S4 Table. Statistical evaluations of each soil abiotic variable and its contribution to the shift observed in the acidobacterial community using the corr.axes command in MOTHUR.**
(PDF)

**S5 Table. Spearman correlation test results for correlations between acidobacterial SDs to soil abiotic properties.** Spearman's rho (ρ) correlation coefficient is given.
(PDF)

## Acknowledgments

The authors of this article would like to thank A. Vorster and C. van Heerden for their help with the sequencing. We would also like to thank CapeNature and the reserve managers of

Jonkershoek and Hottentots Holland (M. Ruthenberg), and Kogelberg (M. Johns) for the opportunity to sample in these pristine locations.

## Author Contributions

**Conceptualization:** Tersia Andrea Conradie, Karin Jacobs.

**Data curation:** Tersia Andrea Conradie.

**Formal analysis:** Tersia Andrea Conradie.

**Funding acquisition:** Karin Jacobs.

**Investigation:** Tersia Andrea Conradie.

**Methodology:** Tersia Andrea Conradie.

**Project administration:** Karin Jacobs.

**Software:** Tersia Andrea Conradie.

**Supervision:** Karin Jacobs.

**Visualization:** Tersia Andrea Conradie.

**Writing – original draft:** Tersia Andrea Conradie.

**Writing – review & editing:** Karin Jacobs.

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
