## [Decision Letter · Decision Letter 0]

17 Feb 2021

PONE-D-21-02341

Distribution patterns of Acidobacteria in different fynbos soils

PLOS ONE

Dear Dr. Jacobs,

Thank you for submitting your manuscript to PLOS ONE. After careful consideration, we feel that it has merit but does not fully meet PLOS ONE’s publication criteria as it currently stands. Therefore, we invite you to submit a revised version of the manuscript that addresses the points raised during the review process.

The manuscript was reviewed by two experts on the ecology and biology of Acidobacteria. As you can see their comments on the current study are very different.

I suggest you respond point-by-point the comments of the reviewers.

We look forward to receiving your revised manuscript.

Kind regards,

Eiko Eurya Kuramae, Ph.D.

Academic Editor

PLOS ONE

Additional Editor Comments:

Dear Dr. Jacobs,

Your manuscript was reviewed by two experts on the ecology and biology of Acidobacteria. As you can see their comments on your study are very different.

I suggest you respond point-by-point the comments of the reviewers.

Journal Requirements:

2. We note that Figure S2 in your submission contain map images which may be copyrighted. All PLOS content is published under the Creative Commons Attribution License (CC BY 4.0), which means that the manuscript, images, and Supporting Information files will be freely available online, and any third party is permitted to access, download, copy, distribute, and use these materials in any way, even commercially, with proper attribution. For these reasons, we cannot publish previously copyrighted maps or satellite images created using proprietary data, such as Google software (Google Maps, Street View, and Earth). For more information, see our copyright guidelines: http://journals.plos.org/plosone/s/licenses-and-copyright.

2.1.    You may seek permission from the original copyright holder of Figure S2 to publish the content specifically under the CC BY 4.0 license. 

2.2.    If you are unable to obtain permission from the original copyright holder to publish these figures under the CC BY 4.0 license or if the copyright holder’s requirements are incompatible with the CC BY 4.0 license, please either i) remove the figure or ii) supply a replacement figure that complies with the CC BY 4.0 license. Please check copyright information on all replacement figures and update the figure caption with source information. If applicable, please specify in the figure caption text when a figure is similar but not identical to the original image and is therefore for illustrative purposes only.

Reviewers' comments:

Reviewer's Responses to Questions

**Comments to the Author**

1. Is the manuscript technically sound, and do the data support the conclusions?

Reviewer #1: Yes

Reviewer #2: Yes

2. Has the statistical analysis been performed appropriately and rigorously? 

Reviewer #1: Yes

Reviewer #2: Yes

3. Have the authors made all data underlying the findings in their manuscript fully available?

Reviewer #1: Yes

Reviewer #2: Yes

4. Is the manuscript presented in an intelligible fashion and written in standard English?

Reviewer #1: Yes

Reviewer #2: Yes

5. Review Comments to the Author

Reviewer #1: In this study, the authors aimed to evaluate the relative abundance and diversity of Acidobacteriota in three fynbos nature reserve soils, trying to link their distribution to soil parameters. The authors observed a high relative abundance of Acidobacteriota in all their different soils and they evaluate further the composition at subdivision level. They discuss the possible roles of the different subdivisions in their soils. The importance is well described, the language and writing are adequate.

Minor issues:

Abstract and elsewhere – Acidobacteriota is the current accepted nomenclature - Whitman WB, Oren A, Chuvochina M, da Costa MS, Garrity GM, Rainey FA, Rossello-Mora R, Schink B, Sutcliffe I, Trujillo ME, et al. Proposal of the suffix -ota to denote phyla. Addendum to 'Proposal to include the rank of phylum in the International Code of Nomenclature of Prokaryotes'. Int J Syst Evol Microbiol 2018; 68:967-969

Line 40 – You do not report new strategies for isolation of new microbes, this sentence is misleading.

Line 58- includes

Line 103 – Were the samples of each type of soil mixed together or each was treated as a replicate? It would be good to have this described.

Line 125 – missing the references of both primers

Line 169-174 – There is no need to repeat the numbers that are in the table.

Line 333 – where, instead of were

Reviewer #2: The manuscript present data on the relative abundances of Acidobacteria subdivisions and their correlations with abiotic factors. The study was well conducted, but the type of the study and the information obtained are not completely new, except for the area studied.

The text is well written and I only have few suggestions:

Line 54. Could the authors find another word for "proliferation"? In microbiology this word implies growth.

Line 49. (…) 16S rRNA gene (as properly written in line 186).

Line 77. It would be nice to state the relative abundance of Acidobacteria previously found in Fynbos habitats (there are many studies cited here, and this information can only be found in the discussion section).

Line 208. 55% of the community or of the Acidobacteria community?

The Discussion needs some rewriting, it is mostly repletion of the results text.

In the discussion section of SD2 (lines 294- 320), the authors may consider add the information in reference: Catão, Elisa CP, et al. " International journal of microbiology 2014 (2014). They did a similar discussion on subgroup 2

6. PLOS authors have the option to publish the peer review history of their article (what does this mean?). If published, this will include your full peer review and any attached files.

Reviewer #1: No

Reviewer #2: No

---

## [Author Response · Author response to Decision Letter 0]

23 Feb 2021

The S2 Figure has been removed from the manuscript, as suggested by the academic editor, and the manuscript and file names have been checked to ensure it meets the style requirements of PLOS ONE.

A rebuttal letter has been attached with a point-by-point response to the comments of the reviewers.

---

## [Editor Report · Decision Letter 1]

9 Mar 2021

Distribution patterns of Acidobacteriota in different fynbos soils

PONE-D-21-02341R1

Dear Dr. Coradie,

We’re pleased to inform you that your manuscript has been judged scientifically suitable for publication and will be formally accepted for publication once it meets all outstanding technical requirements.

Kind regards,

Eiko Eurya Kuramae, Ph.D.

Academic Editor

PLOS ONE

Additional Editor Comments (optional):

The authors have clarified the comments of the reviewers and corrected the mistakes what changes have improved the manuscript.
---

## [Editor Report · Acceptance letter]

12 Mar 2021

PONE-D-21-02341R1 

Distribution patterns of Acidobacteriota in different fynbos soils 

Dear Dr. Conradie:

I'm pleased to inform you that your manuscript has been deemed suitable for publication in PLOS ONE. Congratulations! Your manuscript is now with our production department. 

Kind regards, 

on behalf of

Dr. Eiko Eurya Kuramae 

Academic Editor

PLOS ONE